# Malaria Risk Drivers in the Brazilian Amazon: Land Use—Land Cover Interactions and Biological Diversity

**DOI:** 10.3390/ijerph20156497

**Published:** 2023-08-01

**Authors:** William Gonzalez Daza, Renata L. Muylaert, Thadeu Sobral-Souza, Victor Lemes Landeiro

**Affiliations:** 1Programa do Pós-Graduação em Ecologia e Conservação da Biodiversidade, Departamento de Biociências, Av. Fernando Corrêa da Costa, 2367, Cuiabá 78060-900, MT, Brazil; 2Molecular Epidemiology and Public Health Laboratory, School of Veterinary Science, Massey University, Palmerston North 4472, New Zealand; renatamuy@gmail.com; 3Departamento de Botânica e Ecologia, Instituto de Biociências, Universidade Federal de Mato Grosso (UFMT), Cuiabá 78060-900, MT, Brazil; thadeusobral@gmail.com (T.S.-S.); vllandeiro@gmail.com (V.L.L.)

**Keywords:** malaria, Amazon biome, INLA, land use/cover interactions, bird and amphibian richness-endemics, landscape composition, biological diversity, spatio-temporal modeling

## Abstract

Malaria is a prevalent disease in several tropical and subtropical regions, including Brazil, where it remains a significant public health concern. Even though there have been substantial efforts to decrease the number of cases, the reoccurrence of epidemics in regions that have been free of cases for many years presents a significant challenge. Due to the multifaceted factors that influence the spread of malaria, influencing malaria risk factors were analyzed through regional outbreak cluster analysis and spatio-temporal models in the Brazilian Amazon, incorporating climate, land use/cover interactions, species richness, and number of endemic birds and amphibians. Results showed that high amphibian and bird richness and endemism correlated with a reduction in malaria risk. The presence of forest had a risk-increasing effect, but it depended on its juxtaposition with anthropic land uses. Biodiversity and landscape composition, rather than forest formation presence alone, modulated malaria risk in the period. Areas with low endemic species diversity and high human activity, predominantly anthropogenic landscapes, posed high malaria risk. This study underscores the importance of considering the broader ecological context in malaria control efforts.

## 1. Introduction

Malaria is a tropical and subtropical endemic disease that affects several countries worldwide. In South America, malaria cases were reduced by 58% during 2000–2020, from 1.5 to 0.65 million cases [1]. Yet, malaria remains an important public health problem, far from elimination in several regions. Venezuela, Brazil, and Colombia make up 77% of all cases in South America, and 68% of those cases are due to the *Plasmodium vivax* parasite. While significant efforts have been made to control malaria in Brazil, it continues to pose a public health concern, particularly due to the growing reintroduction of epidemics in areas that have been free of cases for several decades [2]. 

Even though *Anopheles darlingi* is the predominant malaria vector in Amazonian countries [3], there are 61 *Anopheles* species in Brazil, mostly distributed in the Amazon area belonging to the subgenera *Anopheles*, *Kerteszia*, and *Nyssorhynchus*, some of which inhabit other biomes such as the “Atlantic Forest” and the “Pantanal” wetlands, of which 18 species were reported infected with *Plasmodium* parasites; while the highest vector species richness is located in the Amazon biome, with *Nyssorhynchus* and *Anopheles* being the most diversified subgenera [4]. That way, in Brazil, transmission remains highly clustered in the Amazon basin, with 99.5% of the total cases [5,6]. 

Deforestation plays a crucial role in the rise of malaria cases in the Amazon region [7,8,9]. The link between deforestation and malaria incidence is influenced by multiple factors. For instance, at the interface between primary forest and human settlements [10,11], the rapid adaptation of disease-carrying vectors to newly modified environments [12,13] shifts in the food chain and impacts the population of Anopheles mosquitoes, causing the decline of Anopheles predators [14,15].

Deforestation arises from expanding urban infrastructure, agricultural practices, livestock farming, mining, and other human activities. These developments increase the interface between human settlements and natural land cover types, where potential disease vectors reside. Consequently, there is a risk that these vectors, having come into contact with humans infected with malaria, could subsequently infect other people. Thus, spatial modeling of land use and cover is considered an important tool to understand how malaria increases or decreases and offers valuable insights into causal relationships due to the distribution of people and changes in vector habitat quality [16,17,18]. The significance of incorporating landscape-based approaches, such as landscape configuration, into eco-epidemiological models is noteworthy. This goes beyond solely considering the quantity of each category and extends to encompass the interactions between different landscape elements [19]. 

Understanding the role of biodiversity in malaria prevalence, particularly in relation to the dilution effect, is crucial alongside acknowledging the impact of deforestation. By exploring the connections between predator abundance, competitive species, and non-competitive hosts, we can gain insights into how biodiversity influences disease transmission dynamics, such as increased predator populations that could suppress disease-carrying vectors or the presence of non-competitive hosts; thus, reducing malaria transmission [20].

The effects of human-induced changes and biodiversity reduction impacts on disease prevalence have already been studied for hantavirus [21,22], Lyme disease [23], and schistosomiasis [24]. Consequently, in diverse ecosystems, the transmission of diseases is disrupted due to a dilution of infectious agents across different species. In other words, if there are more species in an ecosystem, there are more opportunities for a pathogen to infect a non-host species (interrupting the pathogen cycle), which reduces the concentration of the pathogen in the environment and, consequently, its transmission to the host species [25]. Overall, the dilution effect highlights the importance of preserving biodiversity as a means of reducing the incidence of infectious diseases [26].

Several predators can significantly control populations of disease-carrying mosquitoes in both aquatic and terrestrial ecosystems. The aquatic predators of mosquitoes are composed of different taxa, such as tadpoles, but a few other species have also been identified as effective predators [15]. For example, fish [27] have been identified as very effective predators, particularly *Poecilia reticulata* and *Gambussia affinis*, with the most important group being macroinvertebrates such as Belostomatidae, Notonectidae, and Odonata [28,29]. The most common and effective terrestrial predators are Arachnyda, in addition to mammals such as insectivorous bats [30] and birds such as the flycatcher [31]. Although the majority of groups that regulate mosquito populations have been briefly discussed, it is worth noting the connectivity of a pristine ecosystem. Ecosystem disruptions can lead to an ecological imbalance resulting in a decline in the ecosystem’s ability to control disease-carrying mosquito populations.

Three main factors determine the distribution and biological cycle of malaria: (1) anthropological factors such as migrations, economic activities (hunting, fishing, and agriculture), race, age, and gender; (2) environmental factors such as temperature, relative humidity, altitude, and precipitation [32]; (3) ecological factors such as vegetation types, biological interactions, and nutrient availability, among others. Previous studies evaluated biological factors, land use–land cover (hereafter LULC) influence [33,34,35,36], and malaria spatial patterns [37]. However, the combined impacts of land use, biodiversity, and environmental factors on a macroecological scale (biome) have not been studied yet in Brazil.

Here, we examined malaria cases and the impact of LULC, local biodiversity, and other environmental factors as drivers of malaria risk in the Amazon. Specifically, we aimed to understand: (i) how land use types can increase or decrease malaria risk, (ii) how the combination or interaction of different land use types can affect malaria risk, (iii) whether biodiversity would dilute malaria in higher species number regions, and (iv) how climatic variables correlate with malaria. We expected that Amazonian municipalities with high levels of anthropogenic land use (high habitat modification) and low biological richness had higher malaria prevalence. This information will contribute to the understanding of the spatio-temporal dynamics of malaria across Amazonia, in addition to improving regional and municipal prevention plans directing efforts towards areas characterized by specific landscape, climate, and biodiversity patterns that amplify the risk of malaria.

## 2. Materials and Methods

### 2.1. Dataset

This study was conducted using data for the Brazilian municipalities within the Amazon boundaries, where municipality polygons acted as our sampling units (see Appendix A). The Amazon boundaries used here were taken from a map provided by the Instituto Brasileiro de Geografia e Estatística [38]. 

### 2.2. Malaria Cases and Population Data

Human annual malaria cases (infection location) for each municipality from 2007 to 2018 were provided by Brazil’s Epidemiological Surveillance Information System for Malaria [39]. Due to the low amount of *P. ovale* and *P. malariae* (mostly African and Asian distribution) cases, the models were restricted to *P. vivax* and *P. falciparum*. Mixed-infection cases (coinfections with *P. falciparum* and *P. vivax)* were added to both *falciparum* and *vivax* cases. The cases were downloaded without stratification by age, sex, or race. The Annual Parasite Index (API) was calculated as annual cases/population × 1000. The total population size was downloaded directly from the Instituto Brasileiro de Geografia e Estatística [40]. 

### 2.3. Land Use–Land Cover (LULC)

LULC rasters with 100 m cell-size resolution were downloaded from “Projeto MapBiomas” collection 5.0 to reconstruct landscape structure annual information for 2007–2018, based on Landsat images [41]. Data were extracted through Google Earth Engine, with 13 LULC classes: forest plantation, mining, sugar cane, wetland, temporary crops, grassland, other non-forest formation, savanna, river, lake and ocean, urban infrastructure, pasture, and forest formation. We obtained the area of each LULC classification for each municipality in hectares from 2007 to 2018 using zonal statistics in QGIS 3.8.2. Then, the LULC areas were divided by the municipality area in order to calculate the proportion of each LULC class for each municipality. Finally, in order to characterize the municipalities and their land use change dynamics throughout the years analyzed, the % change in the most important uses in terms of anthropic activities and natural habitats were plotted on maps. 

### 2.4. Environmental Variables 

We assigned a climatic zone to each municipality to establish the four rainiest months, the four driest months, the four warmest months, and finally, the four coldest months. We used two climatic variables from the historical monthly weather data from Worldclim from the years 2007 to 2018 with a spatial resolution of 2.5 min (pixel area ~21 km^2^), the average maximum temperature (°C), and monthly precipitation (mm) [42]. Each climatic zone is formed by a 250,000 km^2^ grid = 500 km × 500 km (see Appendix A). 

Within each climatic zone, we observed rainfall and monthly temperature, specifically focusing on the driest and rainiest four months. If the centroid of a municipality fell within a climatic zone, we assigned the dry or rainy season based on the observed pattern of corresponding months in that climatic zone (Appendix A). After defining the dry and rainy seasons (each season composed of four months), we calculated the total rainfall and average temperature values. Thus, we obtained the following four climatic variables: (1) total rainfall in the dry season, (2) total rainfall in the rainy season, (3) maximum temperature in the dry season, and (4) maximum temperature in the rainy season. The seasons observed were confirmed using the rainfall trends in the Amazon from the past eight decades [43]. Finally, the last environmental variable measured was the mean municipality altitude extracted from EarthEnv with 1 km^2^ cell-size resolution [44]. 

### 2.5. Biological Diversity Variables 

We used data from birds and amphibian species richness and endemism as biological diversity proxy. We used them due to high-quality data availability for these three groups commonly used in macroecological studies [45,46,47]. These data were extracted from the Patterns of Vertebrate Diversity and Protection in Brazil database [48]. The bird and amphibian variables were fixed for all the years studied due to the absence of yearly information and represented the zonal average by the municipality to make comparisons with the other variables and their possible impact on malaria risk.

### 2.6. Model Building

After the variable transformation (common logarithm), we analyzed variable distribution to determine if they have a good representation across the study region. Skewed variables were eliminated from the analysis after graphical method confirmation through the histograms and normality plots (see Appendix A). We also performed a Spearman correlation analysis (*p* < 0.05 of significance) (see Appendix A), finding the most correlated variables (>0.8 of correlation) and eliminating them from the analysis based on the most appropriate biological criteria (such as the elimination of savanna due to the high ecological similarity with grassland or the high correlation of mammal’s richness/endemics with amphibians and birds). The final selected variables are shown in Table 1. 

We initially used cluster analysis to identify the spatial pattern of malaria (random, aggregate, or uniform). Then, we utilized integrated nested Laplace approximation (INLA) analysis to validate the risk factors (LULC, environmental variables and local diversity) driving malaria risk. Cluster analysis is a technique used to identify groups or observations in a dataset based on the similarity of their attributes. In disease spatial analysis, cluster analysis can be used to identify spatial areas with a high prevalence or incidence of a particular disease. INLA is a Bayesian method used for fitting models to spatial data. It is particularly useful for spatial analysis because it can handle complex spatial structures, including spatial correlation and time dependence, and can provide accurate estimates of uncertainty. In disease spatial analysis, INLA can be used to model the spatial distribution of a disease, taking into account the underlying spatial structure of the data and any potential confounding variable [49]. Using cluster analysis and INLA in parallel can provide a more comprehensive analysis of disease spatial patterns.

The cluster analysis models were developed using a retrospective spatiotemporal model under permutation probability using a maximal cluster size of 50% of the total annual municipality population using SaTScan™. The null hypothesis of no clustering was rejected when the simulated *p*-value was lower than or equal to 0.05. For the INLA analysis, we used two models for *P. vivax* and two for *P. falciparum*, covariates models (see Table 1) and interactions models with the landscape configuration (the land use combinations present in a defined area). 

To address the issue of highly uneven population distribution across the Amazonian municipalities, the malaria cases were standardized by utilizing the function ‘expected’ (expected cases) before incorporating them into INLA models, based on the population size and the observed cases of each municipality and each year and with spatial interpolation techniques to smooth the observed incidence rates and generate a continuous surface of expected rates [50]. Accordingly, we performed two models for each parasite, the first one with the covariates described in Table 1 and the second one with the LULC interactions; that is, the relationship between multiple land use/land cover types and their combined effect on malaria risk. Once we determined the influence of each combination of LULC on malaria risk through the INLA models, we created a 10 km × 10 km mesh with each cell representing a specific geographic area. Using this mesh, we constructed a zonal histogram to identify the LULC combinations present in each cell for each year. Finally, we created annual color-coded maps showing the associated malaria risk values for each LULC combination across the study area.

All the models were performed with non-informative priors. We included independent and identically distributed (iid) random effects often used to account for overdispersion [51] and a Besag–York–Mollier spatial term where the observations and data in contiguous municipalities may be spatially correlated compared to areas that are further apart [50,52]. 

The model choice was based on the fit and adequacy, the cross-validation checks via conditional predictive ordinate (CPO) values, the log-likelihood, the dispersion, and the correlation between the mean fitted values and the observed cases. All analyses were performed using the R language (R Core Team, 2022). Correlation analysis was performed using the R package Vegan [53]; the expected values were calculated with the SpatialEpi R package and the neighborhood matrix with the spdep R package. For the spatiotemporal models, we used the R package INLA 21.11.22 [49,54], and the coefficient plots were created using the coefINLA R package [55]; for the maps, zonal statistics, and zonal histograms, we used QGIS 3.8.2 (2018).

## 3. Results

### 3.1. Malaria Cases

In total, 2,827,546 cases were reported. *P. vivax* cases were the most abundant (85.2% of cases), followed by *P. falciparum* (13.7% of cases) and mixed forms (0.9% of cases). In general, during the period 2007–2018, there was a decreasing trend for *P. vivax* and *P. falciparum* cases. The northern zone in the Amazon region concentrated the largest quantity of cases (Appendix A) for both *P. vivax* and *P. falciparum*. In 2007, the Amazonas state (AM) registered the highest number of cases (total cases), 203,164 cases (7.1% of the total cases), which represented the maximum peak of cases among all states for all years from 2007 to 2018.

However, there were areas with absence or rare cases where the relative risk maps showed positive risk (e.g., Codajás—AM, Altamira—PA, and Oriximiná—PA municipalities). Another region that presented high API and relative risk was the northern area in the Amapá state, a state composed mainly of forest cover, grassland, and a big river cover due to the mouth of the Amazon River. Over the years, the API relative risk in the northwest Amazon region has consistently increased. This area is mainly characterized by grasslands and pastures that may or may not be grazed and forms the ecotone between the savannahs of the Orinocense plains and the Amazon biome. Additionally, this region has experienced significant natural habitat modification in the past two decades (see Appendix A). Regarding *P. falciparum*, the pattern remains quite similar, albeit with fewer risk values and smaller clusters. The Amazon’s northwestern and northeastern regions also exhibited elevated API values and positive risk values (Appendix A).

### 3.2. Spatial Clusters

According to our analysis, we identified three distinct clusters for both *P. vivax* and *P. falciparum*, each with varying levels of risk, geographic coverage, time windows, and location (see Table 2). In addition to presenting differences in cluster size, they all had different years with the exception of cluster number 3 for both parasites (from 2007 to 2008). The clusters associated with *P. vivax* had a larger number of expected and observed cases, indicating a higher incidence of this strain of the disease in the affected areas. Additionally, our findings revealed that cluster number 2 for falciparum malaria had a consistent location within regions that experienced notable natural cover transformation over the past two decades, specifically in the Para state (Northeast Amazon zone).

### 3.3. LULC Change 

During 2007–2018 there was notable land use modification (Figure 1). The forest formation in the northern area of Mato Grosso, as well as in nearly all of Acre, Rondonia, and Amazonas states, showed a decline. Porto Velho—RO had one of the highest negative trends in all of the Amazon region. In 40 municipalities, there was a reduction in approximately 10% to 22% of forest cover in the period. Grasslands had a cover reduction as well, but differently from forest formation, the higher percentage change was in the southwest and southeast parts of the Amazon region (Mato Grosso, Tocantins, and Maranhão states). Temporary crops had a notable increase within the agriculturally expansive states (Mato Grosso, Maranhão, and Pará). We want to remark that changes in percentages within larger municipalities correspond to a greater number of hectares, as the analyses were conducted considering the proportion of each land use (land use area/municipality total area × 100). 

### 3.4. Diversity Variables 

Based on the zone statistics (mean by municipality for the variables of richness and endemic species), the states of Acre, western Amazonas, and northern Roraima contained the highest values of bird richness. In the state of Mato Grosso, the southern region also showed a high number of bird species due to the strong influence of the Pantanal wetlands. Para and eastern Maranhão showed high mean endemic bird species numbers; the other states showed smaller numbers of endemic species and zero values for Acre and some municipalities of Amazonas, Roraima, and Amapá. Finally, the Tocantins state exhibited the highest number of endemic amphibians (Figure 1).

### 3.5. Covariate Significance

After identifying over-dispersion in both *P. vivax* and *P. falciparum* cases, we opted to employ zero-inflated negative binomial error distribution for the covariates model and the negative binomial family for the interactions models. For the *P. vivax* covariate model, the cpo values ranged from 0 to 0.99 with a mean of 0.43. For the *P. falciparum* covariate model, the cpo values ranged from 0 to 0.99 with a mean of 0.62. Appendix A, and Figure 2 show the average and range values for each covariate. There was a difference in the effects of each covariate for *P. vivax* and *P. falciparum* risk. Forest formation had an increasing effect on the relative risk for both *Plasmodium* species. Endemic amphibians, endemic birds, grassland, pasture, temporary crops, and urban infrastructure had a decreasing effect on the relative risk of *P. vivax*. For *P. falciparum*, endemic birds, pasture, temporary crops, and urban infrastructure had a decreasing effect on the relative risk.

### 3.6. Interactions Models and Effect Maps

We found 50 unique landscape configuration combinations from 2007 to 2018 (see the example for 2018 in Appendix A). In the INLA interaction models, we estimated the mean effect for every possible combination (57 combinations for the six LULC categories used), of which we found four positive and seven negative significant landscape configurations for the *P. vivax* model and one positive and one negative significant configuration for the P. falciparum model (see Appendix A). For P. vivax, the landscape configuration that showed the highest average effect was (1) Grassland * River Lake and Ocean, (2) Pasture * Grassland * Temporary crops, (3) Forest Formation * River Lake and Ocean, and (4) Pasture * Forest Formation * Grassland. The other seven combinations all had a decreasing effect on estimated risk, with the strongest negative effect being for Forest formation * Urban infrastructure. In the case of P. falciparum, we found only two landscape combinations that were significant. This is the main reason why we did not make maps representing the average effect in each 10 × 10 square since we would have only two colors, which would not be informative: (1) Grassland * River Lake and Ocean with increasing risk effect and (2) Forest Formation * Grassland * River Lake and Ocean with negative mean effect. All of these interaction effects and single land use effects were mapped for each year (see Figure 3) for *P. vivax* in order to see in more detail each effect on the spatial interaction pattern in all the analyzed years. For P. falciparum, the cpo values ranged from 0 to 0.99 with a mean of 0.51; in the case of P. vivax, the values ranged from 0 to 0.98 with a mean of 0.31.

## 4. Discussion

Our findings suggest that the risk of malaria is influenced by a complex interaction of ecological and anthropogenic factors that determine non-random distribution. One notable factor is the richness and endemism of amphibians, mammals, and birds, which appears to reduce the risk of malaria. Additionally, the presence of forests was associated with increased malaria risk, but this relationship was dependent on contact with anthropic land uses. The covers of anthropic activities were found to reduce malaria risk, possibly due to vector habitat loss and the homogenization of the landscape. However, when these land use covers came into contact with other land uses such as water cover and natural habitats of the vector, the effect was the opposite, and the risk of malaria increased (such as pasture together with grassland and temporary crops that each one of them and independently had a reduction effect in the cases of malaria, which in the interaction models showed an effect of increasing the risk of malaria).

The detected clusters suggest that malaria case distribution was far from random and displayed the same trend with past investigations showing high malaria rates in the municipalities Mâncio Lima—AC, Rodrigues Alves—AC, and Cruzeiro do Sul—AC [37] and the other municipalities that make up each of the clusters, mainly the Amazonas, Northern Acre, and the western zone of Para states. Cluster analysis and Bayesian analysis present similarities in the spatial pattern, where the largest clusters contain municipalities with the highest risk values due to highly suitable conditions for malaria transmission. 

Of the 53 Brazilian municipalities located on the border with Peru, Colombia, Venezuela, Suriname, Guyana, and French Guiana, 39 presented high-risk values consistent with the investigation of [4], which reviewed the spatialization of malaria in Brazil. This reaffirms the importance of carrying out early control and prevention alerts, such as border and tri-border control efforts between Peru, Colombia, and Brazil. Due to the combination of high migration rates and their remote locations from larger health centers and political, administrative centers, it is anticipated that these municipalities will be the most affected areas in the future. Various Brazilian government efforts such as the distribution of bed nets, indoor residual spraying (IRS), early diagnosis and treatment, health education and community engagement, and malaria-vector surveillance and monitoring have been proposed to combat this.

*P. vivax* and *P. falciparum* spatial patterns were differentiated. Those differences are still not completely understood and might be due to multiple reasons. For instance, the Duffy gene is randomly distributed in the population and more frequently found in Afro-descendants from the West African region [56]. The difference in the *Plasmodium* species niches [4], the *Plasmodium* species’ time of gametocyte production, and their lifespan [57] or relapses is a blood-stage infection only observed in *P. vivax* and *P. ovale*, arising from the activation of hypnozoites after the primary infection [58]. On the other hand, the Brazilian government launched a campaign in 2003 for the prevention and control of malaria (acronym in Portuguese PNCM) where its process and result indicators are specifically the percentage of *P. falciparum* cases, autochthonous cases of *P. falciparum*, and parasitic index falciparum annual (IFA), among others concerning this differential control effort [59]. Finally, the presence of significantly expanded regions is likely a result of incomplete treatment usage, as well as the rising resistance of both *Plasmodium* to treatments and mosquitoes to insecticides. 

Although climatic variables are critical in driving malaria transmission risk at local scales, no detectable effect was found for our dataset at the municipality level for explaining annual risk. At the biome level, the spatial and temporal scale at which the climatic variables were measured can affect the inferred malaria cases–climatic influence [60]. Despite inter-annual variations in precipitation and temperature, it could be possibly inadequate in influencing the risk of malaria at the regional scale. Instead, it is suggested that the occurrence of malaria cases is more likely influenced by intra-annual variations, which should be studied at a higher resolution by analyzing climatic variables on a monthly or two-month basis [61]. Furthermore, the differential sizes of municipalities, with some being considerably large, pose challenges in representing their climatic spatial variation with a single metric, consequently diminishing their significant impact on malaria risk. Although microscale environmental characteristics are lost when we aggregate data for a year at the municipality level (such as the differences in the land use temperature), biome-scale models are a fundamental tool because epidemiological data are frequently reported at the administrative regions [62]. In addition, the primary socio-ecological processes that lead to the rise and resurgence of zoonotic diseases may take place on a biome-scale level [63,64,65].

Anthropic land uses such as pasture, temporary crops, and urban infrastructure had a reducing effect on malaria transmission risk. However, in the interaction models, various landscape combinations had an increasing effect on malaria transmission where those anthropic land uses were present (e.g., Pasture * Forest formation * Grassland in *P. vivax*), thus corroborating that human activities such as cattle ranching, grazing, or mining (not analyzed in this model but with much existing information on the positive influence on malaria infections) by themselves do not represent all the risk, but rather, the contact zones and landscape configuration of the areas where the main vectors inhabit; hence, it would be suitable to consider implementing buffer zones between human-driven activities and forested areas to address this concern. Several investigations have shown that deforestation, changes in land use, and anthropic activities are related to malaria transmission in the Amazon, in some cases with increasing effect [13,36,66], in others reducing effect [67], demonstrating that the LULC influence is highly context-specific [68]. 

Although the *Anopheles* species populations per se do not have a perfect correlation with malaria cases, habitat suitability mediated by the environment is a factor that determines the probability of contact between humans and the pathogen [33]. The prediction and habitat suitability maps of the primary vectors of malaria (*A. darlingi* and *A. nuneztovari*) throughout the Amazon biome [34] show areas of high suitability in common with areas of increasing effect on the risk of the maps made from the effect of landscape interactions (e.g., eastern Amazon state), where forest cover or water was always present, there were optimal places for oviposition, and favorable conditions existed for the development and growth of mosquitoes [3]. Despite the adaptability of certain Anopheles species to modified environments [10,16,69], the majority of malaria-transmitting species still depend on the presence of forests for their life cycle [70]. On the other hand, it is important to mention that malaria cases are influenced not only by forest cover but also by the activities on it, such as forest clearance [10,16] and changes in habitat suitability due to forest disturbance [69]. Finally, the areas with more forest cover are the areas with higher deforestation rates [71]. 

Zoonotic disease transmission is also subject to changes in landscape heterogeneity and configuration due to the changes in contact zones, distribution, and availability of vectors, pathogens, and hosts [72]. According to long-term mathematical modeling, malaria population dynamics in developing forest areas show that cases behave as a convex declining curve, the cases increasing until reaching a peak and descending until reaching a lower point close to zero due to the reduction in forest formation and the social incomes related to health facilities [35]. We believe that the effects of landscape interactions are correlated with the heterogeneity of the habitat; in this sense, in cells with a greater number of land uses (greater diversity of habitats) and with the presence of preserved forest, the contact between the infected vector and human greatly increases and is reduced as landscape homogenization occurs with a predominance of anthropogenic land uses. On the other hand, it is worth mentioning that although the reduction in forests is a factor that could affect the populations of some *Anopheles* species, the appearance of vectors adapted to modified environments can play a fundamental role in the transmission of homogenized environments in addition to other diseases mediated by anthropophilic vectors such as dengue, zika, chagas, yellow fever, etc. [73].

In relation to the influence of bird and amphibian richness/endemic species on the prevalence of malaria, the intermediate disturbance hypothesis is something that could support ecosystems with a medium anthropic intervention that presents high values of species richness [74]. As mentioned earlier, it is worth noting that municipalities with significant variations in their spatial characteristics may also demonstrate elevated levels of both malaria cases and species diversity. In other words, areas with high spatial heterogeneity, such as complex landscape configurations, may have a higher likelihood of malaria outbreaks. Accordingly, the relationship between malaria cases and species richness (speaking of predator diversity and non-competent hosts) could be correlative but not causal [64]. Additionally, it is crucial to highlight the limitations associated with the utilization of endemism/richness data, specifically concerning spatial sampling bias and potential errors inherent in spatial interpolation methods. Conversely, it should be noted that remote and underexplored regions may exhibit seemingly “low” values of richness/endemism not necessarily due to inherent ecosystem characteristics but rather due to limitations in standardized sampling efforts. However, several investigations support the hypothesis of a dilution effect with an explanation of the underlying mechanisms that could help to reduce cases. In Central America, the decrease in amphibian populations caused by the pathogenic fungus *Batrachochytrium dendrobatidis* has been found to have a direct correlation with the rise in malaria cases. This aligns with the finding that a higher number of endemic species in an area can reduce the risk of malaria transmission [75]. 

While the exact contribution of birds to human malaria infections remains unclear, the impact of native bird species on malaria cases may be attributed to a comparable mechanism, as observed in the study conducted by Swaddle et al. [26]. Their findings revealed that an increase in bird diversity resulted in a decrease in the incidence of West Nile virus due to the host competition. Moreover, the non-competent host hypothesis could explain the importance of the bird and mammal community. The greater the diversity, the larger the number of non-competitive hosts for various types of pathogens [76]. The bird richness was highly correlated with the mammal richness, and it is possible that mammals also play a fundamental role in malaria control, e.g., the high diversity of warm-blooded hosts decreased malaria cases in the Atlantic Forest in Brazil [20], demonstrating the importance of the availability of several hosts to reduce the probability of infection of humans by being bitten by infected mosquitoes. 

On the other hand, taking the avian malaria transmission system as a model, it is suggested that the structure of the bird community and the characteristics of the landscape are determining factors in explaining the prevalence of *Plasmodium* [77]. However, the conclusions from studies up to date are scale-dependent, with diverse host communities commonly inhibiting the spread of parasites on small scales but following a hump-shaped nonlinear relationship [78]. Our findings suggest that there is an effect of endemic birds and amphibians on the incidence of malaria in humans. This highlights the importance of identifying and analyzing the complex relationships among biological factors influencing the spread of this disease. However, understanding the patterns of the assemblage composition of the possible hosts (community structure, that is, a step beyond richness) through time is crucial to predict where and specifically how specific relationships between diversity and disease occur in natural systems [79].

Finally, it is important to mention the pitfalls when comparing multiple datasets, such as richness/endemics species maps, WorldClim data, and malaria cases from the SIVEP platform in Brazil. Firstly, differences in data collection methods (for the species maps), spatial resolution (10 km per pixel for species maps vs. 1 km for WolrdClim), and temporal coverage differences across these datasets can introduce inconsistencies and biases. Lastly, confounding factors, such as ecological or socio-economic variables, can influence the observed relationships between species richness, climate, and malaria cases, potentially leading to spurious associations. Careful consideration of these pitfalls is crucial for cross-dataset comparisons to ensure reliable results.

## 5. Conclusions

Malaria is part of a complex system that includes several components with specific characteristics, including human and vector population dynamics, imperfect detection, recrudescence, and several other factors. Its patterns occur at multiple scales, such as local, regional, and biome. LULC and interaction analyses at the biome scale indicate that forest formations alone are not responsible for driving most of the transmission risk. The endemic species diversity context, in addition to landscape composition, modulates overall risk. Areas with humans living close to the forest, mostly areas with active deforestation, predominantly anthropogenic landscapes, and with smaller values of richness and endemic species, represent the typical pathogenic landscape of high-risk areas. It is essential to plan long-term sustainable development such as ecotourism, agroforestry, sustainable forestry, sustainable fisheries, and renewable energy. Those actions should occur without abruptly modifying ecosystem cycles and their components. The bird and amphibian diversity patterns suggest a potential effect in the case reduction. We recommend future research with time replicates and community composition analysis to directly test the dilution effect and thus be able to develop evidence-based conservation and nature-based solutions. 

## Figures and Tables

**Figure 1 ijerph-20-06497-f001:**
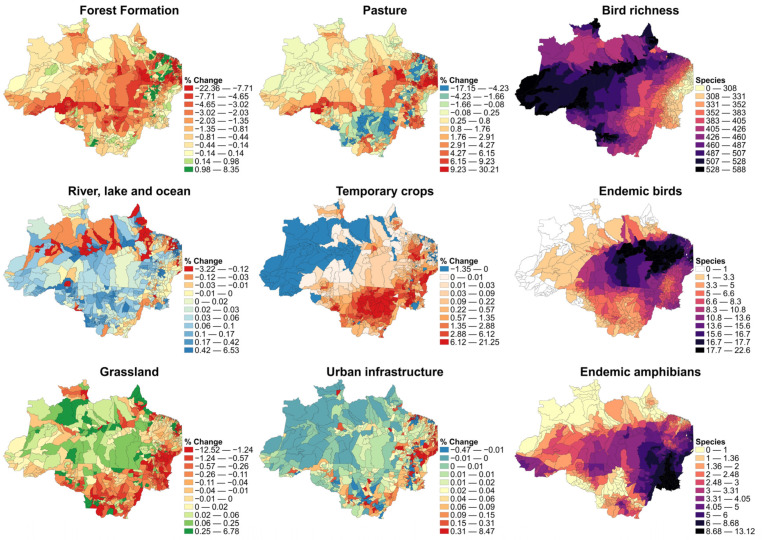
The (**left**) and (**center**) column shows LULC percentage change (%) during the period 2007–2018 at the municipality level; the negative values represent a decrease, and positive values represent an increase in the land use–land cover area of each covariate. The (**right**) column shows the average species richness maps at the municipality level.

**Figure 2 ijerph-20-06497-f002:**
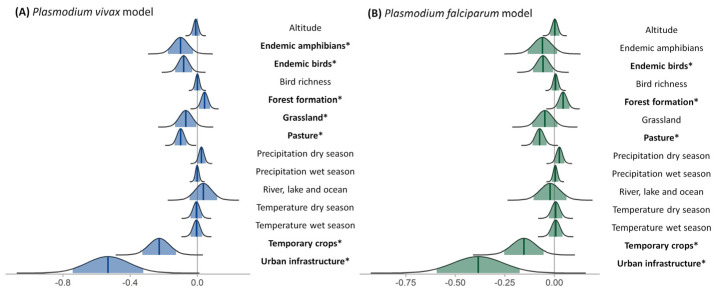
Coefficient plots of posterior distributions effect sizes with the median (dark blue and green lines) and 95% credible intervals (light blue and green shades) of each covariate for (**A**) Plasmodium vivax cases in blue and (**B**) Plasmodium falciparum cases in green; the significant effects are represented in boldface and marked with an asterisk.

**Figure 3 ijerph-20-06497-f003:**
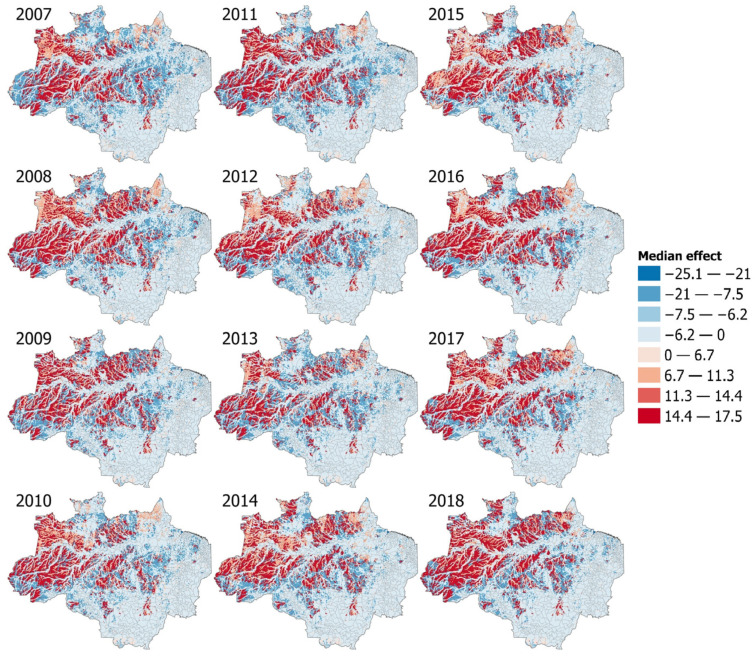
INLA median effect on the credible intervals in the *P. vivax* risk based on the LULC interaction model (100 km^2^ mesh maps). Lower value cells (blue) and higher values cells (red) are based on the landscape configuration (see Appendix A) for all years (2007–2018).

**Table 1 ijerph-20-06497-t001:** Covariate description (values per municipality), including environmental, LULC classifications, and diversity after the correlation analysis and the selection based on the data distribution (see Appendix A). Each of these variables is described for the Amazon biome. A number of total and endemic mammal species were eliminated from the analysis due to the high correlation with birds. * For the details of the species maps methodology for each taxonomic group, see [48].

Variable	Variable Type	Description
Altitude	Topographic	Municipality mean altitude, M.A.M.S.L. (static variable).
Precipitation wet season	Climatic	Total mean precipitation in the wet season (mm).
Precipitation dry season	Climatic	Total mean precipitation in the dry season (mm).
Temperature wet season	Climatic	Mean maximum temperature (°C) in the wet season.
Temperature dry season	Climatic	Mean maximum temperature (°C) in the dry season.
Forest Formation	Land use land cover	Dense rainforest, evergreen seasonal forest, open rainforest, semi-deciduous seasonal forest, deciduous seasonal forest, wooded savanna, and alluvial open rainforest (floodplain forests and Igapó forests) (% of municipality).
Grassland	Land use land cover	Regions within the Amazonia/Cerrado/Orinoco ecotone with a predominance of herbaceous strata (% of municipality).
Pasture	Land use land cover	Area of pasture, predominantly planted, linked to agricultural activity. Areas of natural pasture are predominantly classified as Grassland, which may or may not be grazed (% of municipality).
Temporary crops	Land use land cover	Areas occupied with agricultural crops of short or medium duration, generally with a vegetative cycle of less than one year, which after harvest require new planting to produce, composed mainly of cocoa, rubber, cashew nuts, palm oil, and açaí (% of municipality).
Urban Infrastructure	Land use land cover	Urbanized areas with a predominance of non-vegetated surfaces, including trails, roads, and buildings (% of municipality).
River, lakes and ocean	Land use land cover	As the name denotes, rivers, reservoirs, dams, ocean in the East coast zone in the Amazon region, lakes, and other water bodies (% of municipality).
Endemic amphibians *	Diversity	Mean endemic amphibians species number (static variable).
Endemic birds *	Diversity	Mean endemic bird number species (static variable).
Bird richness *	Diversity	Mean bird number of species (static variable).

**Table 2 ijerph-20-06497-t002:** Results of the cluster analysis for *P. vivax* and *P. falciparum*. The clusters shown here were significant (*p* < 0.00001), and the risk value was calculated by measuring the ratio between observed cases and expected cases. Its geographical extension and location can be observed in Appendix A.

*P. vivax*				*P. falciparum*			
Cluster Time	Observed Cases	Expected Cases	Risk	Cluster Time	Observed Cases	Expected Cases	Risk
(1) 2013–2017	353,973	231,923.58	1.53	(1) 2013–2018	83,331	47,915.42	1.74
(2) 2010–2011	163,179	86,675.19	1.88	(2) 2009–2012	65,919	38,388.91	1.72
(3) 2007–2008	136,239	81,743.78	1.67	(3) 2007–2008	21,794	9446.12	2.31

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
