# Peer review of "Malaria Risk Drivers in the Brazilian Amazon: Land Use—Land Cover Interactions and Biological Diversity"

_ijerph, 2023, doi:10.3390/ijerph20156497_

Round 1

Reviewer 1 Report

The authors have conducted a spatiotemporal analysis of malaria incidence in the Amazon region. It was an interesting paper with a solid approach. I have a few questions and suggestions below.

1.     Abstract line 15 – should be epidemics instead of endemics?

2.     Abstract line 18 – should be “land use/cover interactions, and bird/amphibian richness.”

3.     Abstract – the authors are using the word endemic to describe species abundance. It doesn’t seem quite right, but I am an epidemiologist and so I may be wrong. Are the authors speaking of native birds (contrast to invasive) or some other descriptor? I think it would be fine to say “areas with low species diversity…”

4.     Introduction line 29 – I would drop the word endemic here.

5.     Introduction line 35 – should be epidemic instead of endemic

6.     Introduction lines 37-40. The authors report 61 species of Anopheles in the region. But how many of those are malaria vectors? Not all anophelines can transmit malaria.

7.     Methods lines 180-186. The authors report using a spatial scan statistic to test the question of spatial randomness or clustering. Wouldn’t a Moran’s I be more appropriate here? The spatial scan statistic will tell you where the most likely cluster is, but the Moran’s I will tell you whether there is clustering. (I think you’d get the same result, that there is clustering).

8.     Once the cluster was identified using the spatial scan statistic, it is not clear how the authors incorporated that into the INLA analysis. Was it used, or just done in parallel?

9.     Methods lines 187-189. What are the expected cases that the authors use here? 

10.  I would like to see a figure showing the results of the spatial scan statistic. There are three different spatio-temporal clusters for both species of malaria – how much did they shift over time? That would be quite interesting to display. 

The paper is well written. It could use a review for English language with some rough spots. One specific question I had was around the use of "endemic" to describe the native animals. It was odd to me, but it could be the norm in the authors' field of study.

Author Response

Dear reviewer, thank you very much for the observations, the answers follow point by point. The comments and suggestions have been included in the manuscript, best regards. 

The authors have conducted a spatiotemporal analysis of malaria incidence in the Amazon region. It was an interesting paper with a solid approach. I have a few questions and suggestions below.

  1. Abstract line 15 – should be epidemics instead of endemics?

Thanks for noticing, epidemic by definition is a more appropriate term. Changed as follows: “Despite concerted efforts to maintain control, the reoccurrence of epidemics in regions that have been free of cases for many years presents a significant challenge”.

  1. Abstract line 18 – should be “land use/cover interactions, and bird/amphibian richness.”

No, it is not, since endemic species and species richness are different and both were evaluated in the analysis, but we changed the structure to make the difference clear as follows: “Due to the multifaceted factors that influence the spread of malaria, influencing malaria risk factors were conducted through regional outbreak cluster analysis and a spatio-temporal models in the Brazilian Amazon, incorporating climate, land use/cover interactions, endemic/richness bird, and amphibian species”.

  1. Abstract – the authors are using the word endemic to describe species abundance. It doesn’t seem quite right, but I am an epidemiologist and so I may be wrong. Are the authors speaking of native birds (contrast to invasive) or some other descriptor? I think it would be fine to say “areas with low species diversity…”

When we talk about endemic species, yes, we can describe them as native species both for amphibians and birds, but diversity is a different concept. Diversity measures generally have into count the individual quantity of each species (abundance), but richness and endemic (native) species just count the number of species. We will make that difference clear, thank you so much!

  1. Introduction line 29 – I would drop the word endemic here.

Agree. Thank you, modified as follows: “Malaria is a tropical and subtropical endemic disease that affects several countries worldwide.”

  1. Introduction line 35 – should be epidemic instead of endemic

Thank you, modified as follows: “While significant efforts have been made to control malaria in Brazil, it continues to pose a public health concern, particularly due to the growing reintroduction of epidemics in are-as that had been free of cases for several decades”.

  1. Introduction lines 37-40. The authors report 61 species of Anopheles in the region. But how many of those are malaria vectors? Not all anophelines can transmit malaria.

Thank you, we have check in that article cited (Carlos, B.C., Rona, L.D.P., Christophides, G.K., Souza-Neto, J.A., 2019. A comprehensive analysis of malaria transmission in Brazil. Pathogens and Global Health 113, 1–13.) they reported in the supplementary materials how many species were reported with the Plasmodium parasite, the text were modified as follows: “Even though Anopheles darlingi is the predominant malaria vector in the Amazonian countries [3], there are 61 Anopheles species in Brazil mostly distributed in the amazon area belonging to the subgenus Anopheles, Kerteszia and Nyssorhynchus, some of those species inhabits other biomes such as the “Atlantic Forest” and the “Pantanal” wetlands, of which 18 species were reported infected with Plasmodium parasites” 

  1. Methods lines 180-186. The authors report using a spatial scan statistic to test the question of spatial randomness or clustering. Wouldn’t a Moran’s I be more appropriate here? The spatial scan statistic will tell you where the most likely cluster is, but the Moran’s I will tell you whether there is clustering. (I think you’d get the same result, that there is clustering).

That's right, unlike Moran's I analysis, we wanted to see in which places the clusters were located (see supplementary material #7), if there were several years for the same cluster, and also if those same locations of high degree of grouping had a location in common with the risk maps made with the INLA.

  1. Once the cluster was identified using the spatial scan statistic, it is not clear how the authors incorporated that into the INLA analysis. Was it used, or just done in parallel?

Was done in parallel, as we mentioned before, just to see the cluster spatial pattern common areas with the INLA analysis, we modified the text to make that clear, Thank you. Text modified as follows: “Using both cluster analysis and INLA in parallel can provide a more comprehensive analysis of disease spatial patterns”. 

  1. Methods lines 187-189. What are the expected cases that the authors use here? 

The expected cases are calculated from the population size of each year and each municipality, through the expected function in the Spatial Epi R package. This calculus has also into account population stratification (we don’t use any population stratification). The objective of the expected cases is to balance or standardize the population size differences over the years and in all the municipalities. We expanded that methodology in the article as follows: “To address the issue of highly uneven population distribution across the Amazonian municipalities the malaria cases were standardized by utilizing the function 'expected' (expected cases) before incorporating them into the INLA models, based on the population size and the observed cases of each municipality and each year and with spatial interpolation techniques to smooth the observed incidence rates and generate a continuous surface of expected rates”

  1. I would like to see a figure showing the results of the spatial scan statistic. There are three different spatio-temporal clusters for both species of malaria – how much did they shift over time? That would be quite interesting to display. 

Thanks! Well, the figure 7 of the Supplementary material shows exactly what you are asking for. (see the figure above)

Reviewer 2 Report

This manuscript describes the assessment of land use and biological diversity as drivers of malaria in the Brazilian Amazon region. The research findings have the potential to be of interest to researchers within the malaria fraternity. Unfortunately, the article requires revision to ensure the discussion points and conclusions drawn are clear to the reader. I would also recommend that before resubmission the article is reviewed by a scientific English editor to ensure the requirements of English scientific writing are met. To assist with the revision, I have listed a few comments below.

 Abstract:

Ln15-16: 

It is not clear what point is being made here. Either rephrase more clearly or delete.

Ln16: Some reason for wanting to assess risk needs to be provided

Ln17: Regional analysis was conducted not developed – please correct

Ln19: Was it a high or low level of endemism/richness that was associated with a reduced malaria risk?

Ln20: It is unclear to me whether a positive effect on risk means an increase or decrease in malaria risk – please clarify

Introductions:

Ln34-36: The point being here is unclear – please rephrase for clarity

Ln38-44: What is the relevance of the 61 Anopheles species? Are they all involved in malaria transmission? It is unclear to me why having a number of different Anopheles species means malaria transmission is clustered in the Amazon basin. 

Please provide clarity.

Ln45-49: This paragraph is very unclear and needs to be rephrased to clearly articulate the point being made.

Ln50: What crops are you referring to here, and how do they differ from agriculture?

Ln52: Vectors in forests are unlikely to be infected with human malaria parasites unless they have been in contact with humans. Please clearly make sure this point.

Ln52-54: I am assuming that you are referring to using retrospective data here – so please explain in detail why it is an important tool.

Ln55-57: The point being made here is unclear. Please rephrase and reconsider the use of the word “remarkable”.

Ln59: Please provide more information on how understanding biodiversity will help with malaria control efforts.

Ln64-66: I do not understand how if there are more species, then the pathogen can infect non-hosts, thereby reducing the pathogen pool. Please expand to provide more clarity.

Ln78: What do you mean by “worth nothing the intricacy of a pristine ecosystem”?

Ln85-87: Were these studies done in Brazil?

Ln90: In the context of this study please explain what “dynamics of malaria” means.

Ln94: I am assuming “rich” here refers to species richness and not economic wealth – please provide clarity.

Ln99: It is unclear to me how the study findings could improve control and prevention interventions. Provide a bit more information

Material and Methods:

Ln102: “This study was conducted using data…” please correct

Ln107: Is the data field “infection” referring to parasite species or just confirming a malaria infection?

Ln110-111: Are mixed infection – coinfections with P. falciparum and P. vivax? If so, please state so.

Ln129: This is not a result but a statement that should be in either the introduction or discussion

Ln130: Please explain in a bit more detail how climatic zones were assigned to each municipality. What criteria were used?

Ln150: Which variables are you referring to here? Please state when the richness and endemism data were collected.

Ln159-160: Please expand on the “biological criteria” used to exclude variables. 

Results

Ln216-217: If there was a decreasing trend then it is obvious that cases were highest in 2007.  

Ln218-220: If Amazonas state contributed 203 164 in 2007, how many cases did it contribute over the rest of the study period?

Ln221-222: Provide names of these areas that had a high predicted risk but had no real cases

Ln233: These are spatiotemporal clusters -right? Also, note that the cluster times differ between P. falciparum and P. vivax – this needs to be stated in the results and explained in the discussion section

Ln245: What does “notable change” mean?

Ln247: Provide a p-value to confirm this significant decline

Ln252: Please explain what temporary crops are in the context of this study

Ln253-255: I do not understand what point is being made here – please rephrase to provide more clarity.

Ln262: I do not understand how you can have information on species richness for Acre but no information on species endemicity – please provide a reason for this.

Discussion

Ln315: Again not sure whether a positive effect on risk means a higher or lower malaria risk

Ln316: Was the risk higher only if the humans lived close to areas where land use enhanced the malaria risk?

Ln321: Are you referring to clusters of malaria cases here?

Ln331: I agree that alerts are important, but interventions like increasing access to treatment or improved vector control are essential in areas of high risk. Can you provide any information on interventions currently used?

Ln334: Does this study show that certain areas will be more affected in the future? Did not see that in the results

Ln341-345: Are you saying that the spatial differences between P falciparum and P vivax may be in part be due to the government's efforts to control P. falciparum? If this is the case, state this clearly and explain why P. falciparum is still present in certain areas. 

Ln346-354: Very strange not to find an association with climate – more discussion is required here to explain this.

Ln360-366: This is a bit of a circular argument. If there was no need to visit these areas for farming/mining etc then they would not be exposed to malaria. So how do you then propose to reduce the malaria risk for those involved in mining for example?

Ln368-369: Would have thought this was obvious as not all Anopheles species transmit malaria.

Ln376-377: Not sure what is meant by "that they need for the forest to their life cycle" – please explain this.  

Ln398: What diversity are you referring to here

The are included in the comments above

Author Response

Dear reviewer, thank you very much for the observations, the answers follow point by point. The comments and suggestions have been included in the manuscript, best regards

 Abstract:

Ln15-16:

It is not clear what point is being made here. Either rephrase more clearly or delete.

Thank you, we modified the sentence as follows: “Even though there have been strong efforts to decrease the number of cases, the reoccurrence of epidemics in regions that have been free of cases for many years presents a significant challenge”

Ln16: Some reason for wanting to assess risk needs to be provided

Thanks, the reason was provided as follows: “Due to the multifaceted factors that influence the spread of malaria, influencing malaria risk factors were analyzed through regional outbreak cluster analysis and spatiotemporal models in the Brazilian Amazon, incorporating climate, land use/cover interactions, endemic/richness bird, and amphibian species”

Ln17: Regional analysis was conducted not developed – please correct

Thank you so much, to improve the idea, the sentence was modified as follows: “Influencing malaria risk factors were analyzed through regional outbreak cluster analysis and a spatiotemporal models in the Brazilian Amazon, incorporating climate, land use/cover interactions, endemic/richness bird, and amphibian species.”

Ln19: Was it a high or low level of endemism/richness that was associated with a reduced malaria risk?

Thanks for noticing, the modified sentence: “Results showed that high amphibian, bird richness, and endemism correlated with a reduction in malaria risk”.

Ln20: It is unclear to me whether a positive effect on risk means an increase or decrease in malaria risk – please clarify

Thanks, here is the modified sentence: “Presence of forest had a risk-increasing effect, but it depended on its juxtaposition with anthropic land uses”.

Introduction:

Ln34-36: The point being here is unclear – please rephrase for clarity

Thanks, here is the paraphrased sentence: “While significant efforts have been made to control malaria in Brazil, it continues to pose a public health concern, particularly due to the growing reintroduction of epidemics in areas that had been free of cases for several decades”.

Ln38-44: What is the relevance of the 61 Anopheles species? Are they all involved in malaria transmission? It is unclear to me why having a number of different Anopheles species means malaria transmission is clustered in the Amazon basin.

Please provide clarity.

Thank you for mentioning it, as reviewer 1 also mentioned, the text was modified and also the number of species that are capable of transmitting malaria was added. The modified text below: “Even though Anopheles darlingi is the predominant malaria vector in the Amazonian countries [3], there are 61 Anopheles species in Brazil mostly distributed in the Amazon area belonging to the subgenus Anopheles, Kerteszia, and Nyssorhynchus, some of those species inhabit other biomes such as the “Atlantic Forest” and the “Pantanal” wetlands, of which 18 species were reported infected with Plasmodium parasites; while the highest vector species richness is located in the Amazon biome…”

Ln45-49: This paragraph is very unclear and needs to be rephrased to clearly articulate the point being made.

Thank you, here is the modified paragraph: “Deforestation plays a crucial role in the rise of malaria cases in the Amazon region [7–9]. The link between deforestation and malaria incidence is influenced by multiple factors. For instance, an interface between primary forest and human settlements [10, 11], the rapid adaptation of disease-carrying vectors to newly modified environments [12, 13], shifts in the food chain that impacts the population of Anopheles mosquitoes, and the decline of Anopheles predators [14, 15]”

Ln50: What crops are you referring to here, and how do they differ from agriculture?

You´re right, there is no difference between crops and agriculture in this part. Thank you, here is the modified sentence: “Deforestation arises from the expanding urban infrastructure, agricultural practices, livestock farming, mining, and various other human activities”. 

Ln52: Vectors in forests are unlikely to be infected with human malaria parasites unless they have been in contact with humans. Please clearly make sure this point.

Agree, thank you, here the paragraph modified: “These developments result in an increased interface between human settlements and natural land cover types, where potential disease vectors reside. Consequently, there is a risk that these vectors, having come into contact with humans infected with malaria, could subsequently infect other people”.

Ln52-54: I am assuming that you are referring to using retrospective data here – so please explain in detail why it is an important tool.

Ok, thanks, here is the modified sentence: “Thus, spatial modeling of land use - cover is considered an important tool to understand how malaria increases or decreases and offer valuable insights into causal relationships due to the distribution of people and changes in vector habitat quality”.

Ln55-57: The point being made here is unclear. Please rephrase and reconsider the use of the word “remarkable”.

Thank you so much, here is the modified text: “The significance of incorporating landscape-based approaches, such as landscape configuration, into eco-epidemiological models is noteworthy. This goes beyond solely considering the quantity of each category and extends to encompass the interactions between different landscape elements”

Ln59: Please provide more information on how understanding biodiversity will help with malaria control efforts.

ok, thanks, here is the paragraph including the reasons: “Understanding the role of biodiversity in malaria prevalence, particularly concerning the dilution effect, is crucial alongside acknowledging the impact of deforestation. By exploring the connections between predator abundance, competitive species, and non-competitive hosts, we can gain insights into how biodiversity influences disease transmission dynamics, such as increased predator populations that could suppress disease-carrying vectors, or the presence of non-competitive hosts thus, reducing malaria transmission”

Ln64-66: I do not understand how if there are more species, then the pathogen can infect non-hosts, thereby reducing the pathogen pool. Please expand to provide more clarity.

Ok, thanks, here is the expanded idea: “In other words, if there are more species in an ecosystem, there are more opportunities for a pathogen to infect a non-host species (interrupting the pathogen cycle), which reduces the concentration of the pathogen in the environment and, consequently, its transmission to the host species”.

Ln78: What do you mean by “worth nothing the intricacy of a pristine ecosystem”?

We want to remark on the connectivity of an ecosystem, specifically on the trophic chain, here is the modified text: “It is worth noting the connectivity of a pristine ecosystem”.

Ln85-87: Were these studies done in Brazil?

Some of them have a broad malaria context that includes Brazil. But as the editor and the other reviewer suggested, we have already confirmed and corrected some of the wrong cites, thanks for noticing that.

Ln90: In the context of this study please explain what “dynamics of malaria” means.

Thanks, to be coherent with our objectives, the dynamics word was changed by the word cases. Here is the modified sentence: “Here we examined the malaria cases and the impact of LULC, local biodiversity, and other environmental factors as drivers of malaria risk at Amazon.”.

Ln94: I am assuming “rich” here refers to species richness and not economic wealth – please provide clarity.

Yes, thanks for the correction, the modified sentence is as follows: “Whether biodiversity would dilute malaria in higher species number regions”.

Ln99: It is unclear to me how the study findings could improve control and prevention interventions. Provide a bit more information

Thank you, here is the expanded information: “This information will contribute to the understanding of the spatiotemporal dynamics of malaria across Amazonia, in addition to improving regional and municipal prevention plans directing efforts towards areas characterized by specific landscape, climate, and biodiversity patterns that amplify the risk of malaria.”

Material and Methods:

Ln102: “This study was conducted using data…” please correct

Thank you, the corrected sentence is as follows: “This study was conducted using data for the Brazilian municipalities within the Amazon boundaries”.

Ln107: Is the data field “infection” referring to parasite species or just confirming a malaria infection?

Not just the malaria confirming test, is the parasite type infection as well. The data structure has two options to use the malaria data, where the patient resides or where the patient was infected, to more precisely determine the influence of the variables on the risk of malaria, the place of infection was used, and what parasite species.

Ln110-111: Are mixed infection – coinfections with P. falciparum and P. vivax? If so, please state so.

Thanks, here is the corrected sentence: “The mixed-infection cases (coinfections with P. falciparum and P. vivax) were added to both falciparum and vivax cases”.

Ln129: This is not a result but a statement that should be in either the introduction or discussion

Thanks for the clarification, we remove this sentence as doesn’t belong to this section and expanded it in the discussion section when we talk about climate influence.

Ln130: Please explain in a bit more detail how climatic zones were assigned to each municipality. What criteria were used?

Thank you, we expanded the information to provide more clarity: “Within each climatic zone, we observed rainfall and monthly temperature, specifically focusing on the driest and rainiest four months. If the centroid of a municipality falls within a climatic zone, we assign the dry or rainy season based on the observed pattern of corresponding months in that climatic zone.”

We hope that in this way it has become clearer.

Ln150: Which variables are you referring to here? Please state when the richness and endemism data were collected.

Thanks for noticing, here is the corrected sentence: “The birds, amphibians, and mammal’s variables were fixed for all the years studied due to the absence of yearly information and represented on maps the zonal average per municipality to make comparisons with the other variables analyzed and its possible impact on malaria risk.”

Ln159-160: Please expand on the “biological criteria” used to exclude variables.

Ok, here is the expanded information: “finding the most correlated variables (>0.8 of correlation) and eliminating them from the analysis based on the most appropriate biological criteria (such as the elimination of savanna due to the high ecological similitude with grassland or the high correlation of mammal’s richness/endemics with amphibians and birds).

Results

Ln216-217: If there was a decreasing trend then it is obvious that cases were highest in 2007. 

Agree, we remove that sentence from the text. Thanks.

Ln218-220: If Amazonas state contributed 203 164 in 2007, how many cases did it contribute over the rest of the study period?

Thank you very much for noticing it, the statistic was wrong, in fact, this value of cases represents 7.1%, the text has already been corrected as follows: “In 2007 the Amazonas state (AM) registered the highest number of cases (total cases), with 203,164 cases which represented the maximum peak of cases among all states for all years from 2007 to 2018”.

Ln221-222: Provide names of these areas that had a high predicted risk but had no real cases

There are several areas to name, but we provide some of the more remarkable areas as follows: “There were areas with absence or rare cases, where the relative risk maps, however, showed positive risk (e.g. Codajás – AM, Altamira – PA and Oriximiná – PA municipalities). Thank you.

Ln233: These are spatiotemporal clusters -right? Also, note that the cluster times differ between P. falciparum and P. vivax – this needs to be stated in the results and explained in the discussion section

Thank you, the expanded information is as follows: “). In addition to having differences in cluster size, they all had different years except for cluster number 3 for both parasites (from 2007 to 2008)”

Ln245: What does “notable change” mean?

Significant alteration, we will reformulate it in the text. Thanks.

Ln247: Provide a p-value to confirm this significant decline

We do not carry out a statistical test for this reason, so we are going to eliminate the word significant. Thank you for the correction.

Ln252: Please explain what temporary crops are in the context of this study

We will expand on table 1 for more clarity, thank you. “Areas occupied with crops of short or medium duration, generally with a vegetative cycle of less than one year, which after harvest require new planting to produce, composed mainly by cocoa, rubber, cashew nuts, palm oil, açaí (% of municipality).” As we couldn’t differentiate each of these crops, we let the exact definition provided by Map Biomas made the classification including land use field supervision. 

Ln253-255: I do not understand what point is being made here – please rephrase to provide more clarity.

Thanks, we have changed the structure and added more information as follows: “Based on the zone statistics, (mean by municipality for the variables of richness and endemic species) The states of Acre, western Amazonas, and northern Roraima contained the highest values of bird richness. In the state of Mato Grosso, the southern region also shows a high number of bird species due to the high influence of the Pantanal wetlands”.

Ln262: I do not understand how you can have information on species richness for Acre but no information on species endemicity – please provide a reason for this.

Thank you for noticing, in fact, it was a mistake, it is not an absence of information but values of zero number of endemic species for Acre. It has already been corrected in the text.

Discussion

Ln315: Again not sure whether a positive effect on risk means a higher or lower malaria risk

Thank you, I already corrected it in the text.

Ln316: Was the risk higher only if the humans lived close to areas where land use enhanced the malaria risk?

Not in fact, in some cases where anthropic land use is present together with some natural cover such as Grassland, River lake and ocean, Forest the risk increases. However to point that out, the paragraph was modified as follows: “However, when these land use covers came into contact with other land uses such as water cover and natural habitats of the vector, the effect was reversed, and the risk of malaria increased (such as pasture together with grassland and temporary crops that each one of them and independently had a reduction effect in the cases of malaria, which in the interaction models showed an effect of increase in the risk of malaria).”

Ln321: Are you referring to clusters of malaria cases here?

Thank you, the complete sentence is corrected as follows: “Cluster analysis and Bayesian analysis present similarity in the spatial pattern, where the largest clusters contain municipalities with highest risk values due to highly suitable conditions for malaria transmission.”

Ln331: I agree that alerts are important, but interventions like increasing access to treatment or improved vector control are essential in areas of high risk. Can you provide any information on interventions currently used?

Ok, we provided more information as follows: “… and finally the Brazilian government efforts such as the Distribution of bed nets, Indoor residual spraying (IRS), Early diagnosis and treatment, Health education and community engagement and malaria-vector surveillance and monitoring.

Ln334: Does this study show that certain areas will be more affected in the future? Did not see that in the results

The objectives set for this study were focused on determining the risk factors in malaria cases, making future projections was not within our objectives or calculated with the results of the INLA risk maps since we would not have how to determine the configuration of the landscape or the amount of future land use (which is possible with the weather).

Ln341-345: Are you saying that the spatial differences between P falciparum and P vivax may be in part be due to the government's efforts to control P. falciparum? If this is the case, state this clearly and explain why P. falciparum is still present in certain areas.

Yes, not the main reason but it can be one of the multiple reasons. We will provide other reasons. Thanks.

The modified text is as follows: “On the other hand, the Brazilian government launched a campaign in 2003 for the prevention and control of malaria (acronym in Portuguese PNCM) where its process and result indicators are specifically the percentage of P. falciparum cases, autochthonous cases of P. falciparum, parasitic index falciparum annual (IFA), among others meaning this differential control effort [59]. Finally, the presence of significantly expanded regions is likely a result of incomplete treatment usage, as well as the rising resistance of both Plasmodium to treatments and mosquitoes to insecticides.

Ln346-354: Very strange not to find an association with climate – more discussion is required here to explain this.

for us and according to the expected results, it was also strange, despite the climatic zones we did not find a significant effect, and we leave here the possible additional reasons that were added to the manuscript. Thank you so much. “Despite the inter-annual variations in precipitation and temperature, it could be possibly inadequate in influencing the risk of malaria at the regional scale. Instead, it is suggested that the occurrence of malaria cases is more likely influenced by intra-annual variations, which should be studied at a higher resolution by analyzing climatic variables on a monthly or two-month basis [61]. Furthermore, the differential sizes of municipalities, with some being considerably large, pose challenges in representing their climate spatial variation with a single metric, consequently diminishing their significant impact on malaria risk”

Ln360-366: This is a bit of a circular argument. If there was no need to visit these areas for farming/mining etc then they would not be exposed to malaria. So how do you then propose to reduce the malaria risk for those involved in mining for example?

At the macro scale in which the malaria risk was analyzed, I would propose making use of buffer zones between the vector's habitat and other anthropic uses. Here is the modified text as follows: “Hence, it would be suitable to consider implementing buffer zones between human-driven activities and forested areas to address this concern”.

Ln368-369: Would have thought this was obvious as not all Anopheles species transmit malaria.

Exactly, besides, not all the bites are infectious even with infected mosquitoes.

Ln376-377: Not sure what is meant by "that they need for the forest to their life cycle" – please explain this. 

Thank you, the modified text is as follows: “Despite the adaptability of certain Anopheles species to modified environments [8, 12, 77], the majority of malaria-transmitting species still depend on the presence of forests for their life cycle [39].”  

Ln398: What diversity are you referring to here

Thank you, the sentence was modified as follows: “About the influence of bird and amphibian’s richness/endemic species on the prevalence of malaria”.

Reviewer 3 Report

In this paper, the authors have gathered statistical data for a large number of cases of malaria and compared this to a number of other parameters that could influence the number of cases. The conclusion that biodiversity and not only forest formation modulates the risk of malaria is important information for future planning of geographical areas and in general the paper is well written when it comes to the composition of the paper, but it might need a bit of English editing.

What I would like to see more of in the paper, is discussion about the potential pitfalls when comparing multiple datasets. The authors could also discuss more about other methods that could potentially be used to increase the knowledge of the field.

The authors have to be a lot more meticulous when it comes to citing references. For example, on  line 83, they refer to reference number 2 for the importance of race, age and gender, but these words are not even mentioned in this reference.

Line 158: They have performed a Spearman correlation, but how have they compensated for multiple correlations?

-

Author Response

Dear reviewer, thank you very much for the observations, the answers follow point by point. The comments and suggestions have been included in the manuscript, best regards.

In this paper, the authors have gathered statistical data for a large number of cases of malaria and compared this to several other parameters that could influence the number of cases. The conclusion that biodiversity and not only forest formation modulates the risk of malaria is important information for future planning of geographical areas and in general the paper is well written when it comes to the composition of the paper, but it might need a bit of English editing.

What I would like to see more of in the paper, is discussion about the potential pitfalls when comparing multiple datasets. The authors could also discuss more about other methods that could potentially be used to increase the knowledge of the field.

Thanks for noticing, in the manuscript, we increased the discussion around the use of different data sets (richness/endemics, WorldClim and malaria cases from SIVEP platform) including in this paragraph:

“Finally, is important to mention the pitfalls when comparing multiple datasets, such as richness/endemics species maps, WorldClim data, and malaria cases from the SIVEP platform in Brazil. Firstly, differences in data collection methods (for the species maps), spatial resolution (10 km per pixel for species maps vs 1 km for WolrdClim), and temporal coverage differences across these datasets can introduce inconsistencies and biases. Lastly, the presence of confounding factors, such as ecological or socio-economic variables, can influence the observed relationships between species richness, climate, and malaria cases, potentially leading to spurious associations. Careful consideration of these pitfalls is crucial when conducting cross-dataset comparisons to ensure reliable and meaningful results.”.

The authors have to be a lot more meticulous when it comes to citing references. For example, on line 83, they refer to reference number 2 for the importance of race, age and gender, but these words are not even mentioned in this reference.

Thank you so much for reading in detail, after the references confirmation, actually in the text we were trying to make a list as follows and not a reference:

 (1) anthropological factors such as migrations, economic activities (hunting, fishing, agriculture), race, age, and gender;

 (2) Environmental factors such as temperature, relative humidity, altitude, precipitation [32] and

 (3) ecological factors such as vegetation types, biological interactions, nutrient availability, among others.

In the new version of the manuscript, the references are in superscript to make a clear difference. Thank you so much.

Line 158: They have performed a Spearman correlation, but how have they compensated for multiple correlations?

Initially, a principal component analysis was performed to see associations or correlations of variables in a multivariate space, however, we considered using Spearman's correlation analysis because we needed a statistical test to perform this elimination of variables taking into account a p-value of significance. Finally, it is necessary to point out that after the preselection from the histograms, similar results were found between the PCA analysis and the Spearman correlation analysis.